# Anti-Inflammatory Effects of Heliangin from Jerusalem Artichoke (*Helianthus tuberosus*) Leaves Might Prevent Atherosclerosis

**DOI:** 10.3390/biom12010091

**Published:** 2022-01-06

**Authors:** Papawee Saiki, Mizuki Yoshihara, Yasuhiro Kawano, Hitoshi Miyazaki, Koyomi Miyazaki

**Affiliations:** 1Cellular and Molecular Biotechnology Research Institute, National Institute of Advance Industrial Science and Technology, Tsukuba 305-8566, Japan; y.kawano@aist.go.jp (Y.K.); k-miyazaki@aist.go.jp (K.M.); 2Graduate School of Life and Environment Sciences, University of Tsukuba, Tsukuba 305-8577, Japan; s201821164@gmail.com (M.Y.); miyazaki.hitoshi.gb@u.tsukuba.ac.jp (H.M.)

**Keywords:** heliangin, Jerusalem artichoke, *Helianthus tuberosus*, atherosclerosis

## Abstract

Atherosclerosis is considered the major cause of cardiovascular and cerebrovascular diseases, which are the leading causes of death worldwide. Excessive nitric oxide production and inflammation result in dysfunctional vascular endothelial cells, which are critically involved in the initiation and progression of atherosclerosis. The present study aimed to identify a bioactive compound from Jerusalem artichoke leaves with anti-inflammatory activity that might prevent atherosclerosis. We isolated bioactive heliangin that inhibited NO production in LPS-induced macrophage-like RAW 264.7 cells. Heliangin suppressed ICAM-1, VCAM-1, E-selectin, and MCP-1 expression, as well as NF-κB and IκBα phosphorylation, in vascular endothelial cells stimulated with TNF-α. These results suggested that heliangin suppresses inflammation by inhibiting excessive NO production in macrophages and the expression of the factors leading to the development of atherosclerosis via the NF-κB signaling pathway in vascular endothelial cells. Therefore, heliangin in Jerusalem artichoke leaves could function in the prevention of atherosclerosis that is associated with heart attacks and strokes.

## 1. Introduction

Cardiovascular diseases (CVDs) are the most prevalent cause of death, accounting for ~31% of all deaths worldwide [1]. Such diseases include ischemic heart disease, heart failure, peripheral arterial disease, stroke, and atherosclerosis, which are major contributors to a reduced quality of life [2,3,4]. Arteries structurally comprise the intima and media that mainly comprise vascular endothelial (VECs). Physical stress and excessive ROS damage VECs in the intima and increase the expression of cell adhesion molecules on the surface of VECs. After adhering to VECs, monocytes and lymphocytes are taken up and subsequently infiltrate the intima. Thereafter, monocytes differentiate into macrophages and are transformed into foam cells. These foam cells secrete proinflammatory cytokines and eventually undergo apoptosis. This process leads to inflammation and progressive atheroma formation [5,6]. Foam cell apoptosis is an important feature of atherosclerotic plaque development [6]. This series of events indicates that anti-inflammatory activity in macrophages is important to prevent atherosclerosis. In macrophages, nitric oxide (NO) is produced by inducible nitric oxide synthase and cytokine stimulation [7]. NO functions in several physiological and pathological processes associated with acute and chronic inflammation [8,9]. Therefore, we evaluated the inhibition of NO production in RAW 264.7 cells that are macrophage-like cell lines, to identify anti-inflammatory compounds that might prevent arteriosclerosis.

Jerusalem artichoke is a sunflower native to North America, and its tubers are consumed worldwide [10]. They comprise a functional nutrient that improves postprandial hyperglycemia and diabetes, as well as exerts antioxidant, anticarcinogenic, and anti-fungistatic effects [11,12,13,14]. In contrast, Jerusalem artichoke leaves are unused resources, and the current understanding of their potential value is limited. Therefore, this study aimed to identify a bioactive compound from Jerusalem artichoke leaves with anti-inflammatory activity that might prevent atherosclerosis. Heliangin inhibits inflammatory responses in lipopolysaccharide (LPS)-treated macrophages via the NF-κB pathway [15]. However, the ability of heliangin to suppress inflammatory responses in VECs and prevent atherosclerosis has not been described.

The inflammation of VECs causes arteriosclerosis. The expression of intercellular adhesion molecule-1 (ICAM-1), vascular cell adhesion protein 1 (VCAM-1), and E-selectin increases in VECs during the initiation and progression of arteriosclerosis. Chemoattractant protein-1 (MCP-1) promotes the migration of monocytes and macrophages into VECs. The cell adhesion molecules L-, P-, and E-selectin are expressed in leukocytes, platelets, and endothelial cells, respectively, during the early phase of inflammation [16,17,18,19]. Monocytes and VECs also express ICAM-1 and VCAM-1 that are adhesion molecules expressed via the nuclear factor κ-light-chain-enhancer of activated B cell (NF-κB) signaling pathway upon stimulation with inflammatory cytokines such as tumor necrosis factor (TNF-α) and interleukin-1 (IL-1) [20,21]. Therefore, the suppression of ICAM-1, VCAM-1, MCP-1, and E-selectin in VECs is also important to prevent atherosclerosis. Here, we explored the anti-inflammatory effects of heliangin from Jerusalem artichoke leaves in VECs stimulated with TNF-α. We measured ICAM-1, VCAM-1, E-selectin, and MCP-1 gene expression in VECs stimulated with TNF-α. Then, we analyzed phosphorylation levels of NF-κB, IκBα, p38 mitogen-activated protein kinase (MAPK), and c-Jun N-terminal kinase (JNK) 1/2 to understand the signaling pathways involved in the anti-inflammation.

## 2. Materials and Methods

### 2.1. Cell Culture

Mouse Abelson leukemia virus-transformed, monocyte-macrophage RAW 264.7 cells (Riken Cell Bank, Tsukuba, Japan) were cultured in Dulbecco’s modified Eagle’s medium (DMEM, Fujifilm Wako Pure Chemical Corporation, Osaka, Japan) containing 10% fetal bovine serum (FBS; Biowest, Tokyo, Japan) at 37 °C in a humidified 5% CO_2_ incubator [22].

Vascular endothelial cells (VECs) derived from porcine aortas (Tsuchiura Meat Cooperative, Tsuchiura, Japan) were cultured in DMEM containing 10% FBS, 100 units/mL penicillin, and 100 μg/mL streptomycin at 37 °C in a humidified 5% CO_2_ incubator. The cells were used between passages 8 and 10 for bioassay. The cells were incubated in DMEM containing 1% FBS for 17 h before starting the experiments [23].

### 2.2. Extraction, Isolation, and Purification of Heliangin

Jerusalem artichoke (*Helianthus tuberosus*) leaves were provided by the Japan Diabetes Research Institute (Nagano, Japan and Aiki Co., Ltd., Ibaraki, Japan). Jerusalem artichoke leaves were dried and extracted by ethanol (Fujifilm Wako Pure Chemical Industries, Ltd., Richmond, VA, USA). Crude ethanol extract was fractionated using Wakogel^®^C-300 and hexane/acetone (3:1 and 2:1) (hexane and acetone were purchased from Fujifilm Wako Pure Chemical Industries, Ltd., Richmond, VA, USA). Then, fractions were again fractionated by a Sep-PaktC18 cartridge (Waters Corporation, Milford, MA, USA) with 10–100% methanol. We searched for anti-inflammatory compounds according to the NO inhibitory effects of the various fractions [24]. The active compounds were isolated and purified by HPLC (JASCO International Co., Ltd., Tokyo, Japan) equipped with a Capcell PAK C18 5 µM, 10 mm i.d. 250 mm column (Osaka Soda Co., Ltd., Osaka, Japan) and eluted with a gradient of 40% methanol with 0.1% formic acid and 100% methanol with 0.1% formic acid at a flow rate of 1.5 mL/min. Heliangin was detected using a UV wavelength detector at 254 nm (JASCO International Co., Ltd., Tokyo, Japan) [22]; then, we purified a fraction by HPLC under the conditions described above. All extracts and heliangin were evaporated and dissolved in ethanol for bioassay.

### 2.3. Analysis by Nuclear Magnetic Resonance Spectroscopy (NMR)

Dried heliangin (10 mg) was exchanged into chloroform-D1 0.03 col.% TMS (Sigma-Aldrich Co., St. Louis, MO, USA). Spectra were determined by one-dimensional (^1^H-NMR and ^13^C-NMR) and two-dimensional correlated spectroscopy (COSY) using a Bruker 500 MHz NMR (Bruker, Billerica, MA, USA) [24].

### 2.4. Mass Spectrometry Analyses

Dried heliangin (20 ppm) was dissolved and diluted in LC/MS grade methanol (Fujifilm Wako Pure Chemical Industries, Ltd., Richmond, VA, USA). The precise mass of heliangin was determined using a Q Exactive Plus Mass Spectrometer (Thermo Fisher Scientific Inc., Waltham, MA, USA). Mass spectrometry (ESI-MS) was performed in the positive mode with a scan range of 130–1900 *m*/*z*. The capillary voltage was set to 10 kV, and the capillary temperature was at 200 °C.

### 2.5. Nitric Oxide Production

We cultured RAW 264.7 cells in DMEM containing 10% FBS at 37 °C in a humidified 5% CO_2_ incubator. Then, we seeded 1.6 × 10^5^ cells into 96-well plates containing DMEM for 24 h under standard conditions. The cells were incubated with samples and lipopolysaccharides from *Escherichia coli* O26:B6 (Sigma-Aldrich Co., St. Louis, MO, USA, LPS) at 1.25 g/mL [24]. After 15 h, the culture medium of activated cells was transferred to a new 96-well plate. Griess reagent (Sigma-Aldrich Co., St. Louis, MO, USA) was added before incubating at 37 °C for 15 min. Absorbance was read at 540 nm using an iMarkTM Microplate Reader (Bio-Rad Laboratories, Inc., Hercules, CA, USA). The percentage of NO production was calculated and compared with LPS group [24].

Cell proliferation was determined using CellTiter 96^®^ AQueous One Solution Cell Proliferation Assays (Promega Corp., Madison, WI, USA), as described by the manufacturer. Aqueous One Solution Reagent was added to cells and incubated at 37 °C for 20 min. The absorbance of proliferating RAW 264.7 cells was measured at 490 nm using the iMarkTM Microplate Reader. The percentage of cell proliferation was calculated and compared with LPS group [22].

### 2.6. Cell Proliferation Assays

We seeded 1.67 × 10^5^ VECs into DMEM containing 1% FBS in 24-well plates under standard conditions for 24 h. Then, we incubated VECs with 1, 5, 15, and 45 µM heliangin for 24 h. The VECs were washed with phosphate-buffered saline (PBS); then, VECs were collected using trypsin-EDTA. VECs were stained with 0.5% trypan blue to measure cell viability (Nacalai Tesque Inc., Kyoto, Japan). Cell viability was determined using a Countess II FL cell counter (Thermo Fisher Scientific Inc., Waltham, MA, USA) [23].

### 2.7. Quantitative Real-Time Polymerase Chain Reaction (qRT-PCR)

We seeded 2.5 × 10^5^ VECs into 60 mm dishes containing DMEM with 1% FBS under standard conditions for 24 h. We analyzed the ability of crude ethanol extract and heliangin to inhibit the stimulation of TNF-α in VECs. VECs were incubated with crude ethanol extract (200 µg/mL) and heliangin (15 µM) for 24 h, and then stimulated with TNF-α (1 ng/mL) for 3 h. We also examined the inhibition of TNF-α in VECs using the p38 MAPK inhibitor, SB203580 (Adipogen Life Sciences, Liestal, Switzerland), and the JNK inhibitor, SP600125 (Fujifilm Wako Pure Chemical Industries, Ltd., Richmond, VA, USA). The VECs were incubated with 10 µM of each inhibitor for 1 h followed by TNF-α (2 ng/mL) for 3 h. Total RNA was extracted from whole cells using ISOGEN Total RNA extraction reagent (Nippon Gene Co., Ltd., Toyama, Japan). Complementary DNA was generated using ReverTra Ace™ qPCR RT Kits (Toyobo Co., Ltd., Osaka, Japan) [25]. Quantitative real-time PCR proceeded using Power SYBR Green Master Mix and an Applied Biosystems 7300 Real-Time PCR System (Thermo Fisher Scientific Inc., Waltham, MA, USA). The (5′ → 3′) primer sequences were purchased from Eurofin Genomics Co., Ltd., Tokyo, Japan) and are listed in Table 1. All data were normalized to that of the internal standard 18S rRNA [26].

### 2.8. Western Blotting

We seeded 2.5 × 10^5^ VECs into 35 mm dishes containing DMEM with 1% FBS under standard conditions for 24 h. First, we investigated the effects of 1 ng/mL TNF-α on phosphorylation levels of NF-κB, IκBα, p38 MAPK, and JNK1/2 in VECs incubated for 10, 30, 60, 120, and 180 min by Western blotting. Then, we investigated the effects of crude ethanol extract and heliangin in TNF-α stimulated VECs. VECs were incubated with 200 µg/mL crude ethanol extract and 15 µM heliangin for 24 h; then, VECs were stimulated with 1 ng/mL TNF-α for 10 min. The VECs were washed twice with PBS and immediately stored frozen at −80 °C. Thereafter, the VECs were thawed and resuspended in lysis buffer (50 mM HEPES pH 7.5, containing 50 mM NaCl, 1 mM EDTA, 50% glycerol, 100 mM NaF, 10 mM sodium pyrophosphate, 1% Triton X-100, 1 mM Na_3_VO_4_, 1 mM PMSF, 1.5 mg/mL antipain, 2.5 mg/mL leupeptin, and 10 mg/mL aprotinin). Lysates were centrifuged at 16,000× *g* at 4 °C for 15 min. Supernatants were collected and Western blotted. The concentrations of proteins in the cell supernatants were assayed using Pierce™ BCA Protein Assay Kits (Thermo Fisher Scientific Inc., Waltham, MA, USA); then, the proteins were separated by 10% SDS-PAGE and transferred to PVDF membranes. Nonspecific protein binding on the membranes was blocked with 5% bovine serum albumin (BSA) in Tris-buffered saline with Tween (TBS-T) at room temperature for 1 h; then, the membranes were incubated overnight with the following primary antibodies: anti-phospho-NF-κB (Ser536) anti-p65, anti-phospho-IκBα (Ser32), anti-phospho-p38 (Thr180/Tyr185) anti-phospho-MAPK (Cell Signaling Technology, Danvers, MA, USA), anti-phospho-JNK 1/2 (Thr183/Tyr185), and monoclonal anti-β-actin (Sigma-Aldrich Co., St. Louis, MO, USA) diluted in in 5% BSA in TBS-T at 4 °C. The membranes were then incubated with HRP-conjugated secondary antibody for 2 h at room temperature. Bands were detected using Immobilon ECL Ultra Western HRP Substrate (Merck KGaA, Darmstadt, Germany), and band intensity was quantified using a C-DiGit™ Blot Scanner (LI-COR Biosciences, Lincoln, NE, USA). The protein levels were normalized to that of the internal control β-actin [23].

### 2.9. Statistical Analysis

Data were statistically analyzed by one-way analysis of variance (ANOVA) with Dunnett and Tukey’s test using EZR software version 1.52 [27]. Values are shown as the mean ± SD. Significant differences are shown as *p*-values (*p* < 0.05, *p* < 0.01, *p* < 0.001).

## 3. Results and Discussion

Nitric oxide is a mediator and regulator of inflammatory responses [28]. We previously searched for anti-inflammatory compounds by screening their ability to inhibit NO production in mice macrophages [24]. Here, we investigated the ability of 10–100% methanol fractions of crude ethanol extracts of Jerusalem artichoke leaves to inhibit NO production. Appendix A shows that 50%, 70%, and 90% methanol fractions significantly inhibited NO production, but 70% and 90% fractions were cytotoxic to RAW 264.7 cells. Therefore, we isolated the 50% methanol fraction using high-performance liquid chromatography (HPLC; Appendix A). Then, we determined 11 fractions to inhibit NO production in RAW 264.7 cells stimulated with LPS. Appendix A shows that the most effective inhibitor was fraction 6. Fraction 6 was purified by HPLC as described above. The structure of fraction 6 was determined by ^1^H-NMR, ^13^C-NMR, and COSY (Appendix A). The chemical shifts of published heliangin [29] and fraction 6 (Table 2) on ^13^C- and ^1^H-NMR were identical, confirming that fraction 6 was heliangin. The structure of heliangin is shown in Figure 1. The carbon numbers in Figure 1 are the same as in Table 2. The formula of heliangin is C_20_H_26_O_6_, and the accurate mass of C_20_H_26_O_6_Na is *m*/*z* 385.1630 [M + Na]^+^. The Q-Exactive mass of isolated heliangin (C_20_H_26_O_6_Na) was *m*/*z* 385.1627 [M + Na]^+^ Δ−0.3 mmu. The NMR and mass spectrometry data further substantiated that fraction 6 was heliangin. Heliangin has been identified in *Bejaranoa semistriata*, *Viguiera eriophora*, and *Viguiera puruana* [30,31,32]. 

We assessed the ability of purified 2, 5, 10, 25, and 50 µM heliangin extracted from Jerusalem artichoke leaves to inhibit NO production. The results showed that 50 µM heliangin was slightly cytotoxic to RAW 264.7 cells, whereas a low concentration was not (Figure 2b). Furthermore, 2, 5, 10, and 25 µM heliangin significantly and concentration-dependently reduced NO production (Figure 2a). 

The bioactivity and cytotoxicity of heliangin have been investigated in Ehrlich ascites tumor cells, as well as in human oral epidermal carcinoma, cervical epithelial cells, and liver carcinoma (hepa59T/VGH) [29,33]. Appropriately increased NO production in VECs suppresses vasoconstriction, platelet adherence and aggregation, and monocyte adherence in arteriosclerosis, whereas excessive amounts in macrophages promote atherosclerosis [34], which is initiated by dysfunctional VECs and progresses to chronic inflammation [35]. Monocytes adhere to VECs and transform into macrophages at the early stage of atherosclerosis [36]. Macrophage-mediated inflammation leads to plaque in blood vessels, which contributes to thrombus formation that enhances atherosclerosis [37]. Cell adhesion molecules are expressed on the surface of VECs by inflammatory cytokine induction. Monocytes adhering to these adhesion molecules are absorbed into vascular walls, where they lead to plaque formation [38,39,40]. It was reported that heliangin at 140, 280, 420, and 560 µM suppressed gene expressions of TNFα, IL-6, iNOS, and COX-2 in LPS-stimulated RAW 264.7 cells. Moreover, heliangin inhibits TNFα, IL-6, NO, and PGE_2_ levels in macrophages cells lysate via the MAPK and NF-κB pathway [15]. Our findings showed that heliangin at 2, 5, 10, and 25 µM significantly and concentration-dependently exerted anti-inflammatory activity by reducing NO overproduction in LPS-stimulated RAW 264.7 cells. This indicates that a low concentration of heliangin also has anti-inflammatory activity in macrophages cells. However, [15] did not describe the ability of heliangin to suppress inflammatory responses in VECs. Therefore, we assessed the effects of purified heliangin on the expression of cell adhesion molecules induced by TNF-a (typical inflammatory cytokine) in VECs.

ICAM-1, VCAM-1, E-selectin, and MCP-1 are biomarkers of endothelial dysfunction in atherosclerosis [23,41,42]. We incubated VECs with 1, 5, 15, and 45 µM heliangin for 24 h, and then assayed cell viability using trypan blue staining. Cell viability was significantly reduced by 45 µM heliangin (Appendix A). Therefore, we examined the anti-inflammatory effects in VECs incubated for 24 h with 15 µM heliangin, followed by incubation with 1 ng/mL TNF-α for 3 h. Total RNA was extracted from the VECs; then, levels of ICAM-1, VCAM-1, E-selectin, and MCP-1 mRNAs were measured using qPCR. Figure 3 shows that crude ethanol extract at 200 µg/mL and heliangin at 15 µM significantly suppressed the elevated expression of these mRNAs. 

Appendix A shows that 10 µM SB20358 (p38 MAPK inhibitor) and 10 µM SP600125 (JNK inhibitor) significantly suppressed the expression of ICAM-1, VCAM-1, E-selectin, and MCP-1. Moreover, TNF-α induces ICAM-1, VCAM-1, E-selectin, and MCP-1 expression via the NF-κB signaling pathway in endothelial cells [43,44,45,46,47]. Therefore, we examined the inhibition of TNF-α induction in VECs via the NF-κB and MAPK signaling pathways. First, we investigated the effects of 1 ng/mL TNF-α on phosphorylation levels of NF-κB, IκBα, p38 MAPK, and JNK1/2 in VECs incubated for 10, 30, 60, 120, and 180 min by Western blotting. Incubation with TNF-α for 10 min produced maximal phosphorylation of these proteins (Appendix A). Therefore, we determined the levels of NF-κB, IκBα, p38 MAPK, and JNK1/2 phosphorylation after induction with TNF-α for 10 min, followed by heliangin incubation. The results showed that heliangin suppressed the levels of p-NF-κB and p-IκBα induced by TNF-α, but could not suppress p38 MAPK and JNK1/2 (Figure 4). Although the MAPK and NF-κB signaling pathways are involved in regulating the function of endothelial cells and might be key factors in the formation of arteriosclerosis [48,49], our results suggested that heliangin could suppress TNF-α effects in vascular endothelial cells via the NF-κB signaling pathway. Notably, NF-κB has been identified in human atherosclerotic plaque [50]. These findings suggested that heliangin exerts anti-inflammatory effects on macrophages and VECs, supporting the notion that it could prevent atherosclerotic plaque and, thus, contribute to the prevention of atherosclerosis. 

## 4. Conclusions

Atherosclerosis causes vascular disease, which is the leading cause of mortality worldwide. Therefore, we screened compounds from Jerusalem artichoke leaves for compounds with anti-inflammatory activity that might prevent atherosclerosis. We isolated a compound with potential anti-inflammatory activities using open column chromatography and purified it by C18 HPLC. We then confirmed that the compound was heliangin by NMR and mass spectrometry. We showed that heliangin inhibited NO production in RAW 264.7 cells induced by LPS and suppressed gene expression of ICAM-1, VCAM-1, E-selectin, and MCP-1 in VECs induced by TNF-α. Heliangin also suppressed NF-κB and IκBα phosphorylation induced by TNF-α in VECs. Therefore, heliangin appeared to suppress ICAM-1, VCAM-1, E-selectin, and MCP-1 induction by TNF-α in VECs via the NF-κB signaling pathway. These results suggest that heliangin derived from Jerusalem artichoke leaves could play a role in the prevention of inflammatory in macrophage and VECs, which might help to prevent atherosclerosis.

## Figures and Tables

**Figure 1 biomolecules-12-00091-f001:**
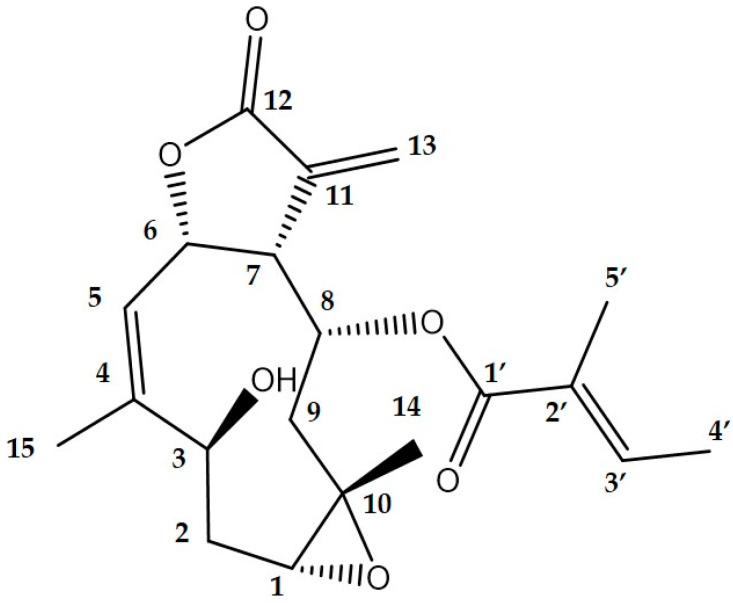
The structure of heliangin.

**Figure 2 biomolecules-12-00091-f002:**
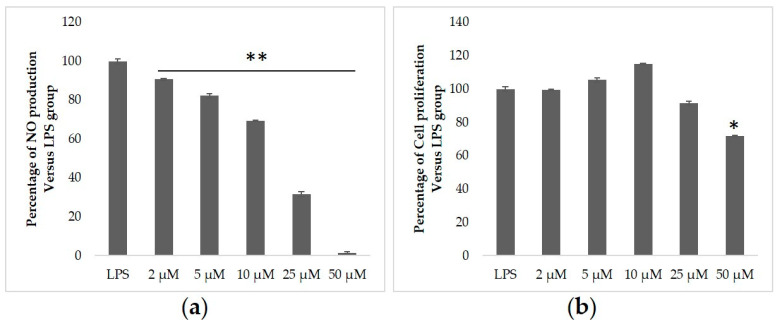
Effects of heliangin from Jerusalem artichoke leaves in RAW 264.7 cells stimulated with LPS. (**a**) Inhibition of NO production. (**b**) Cell proliferation. Values are shown as the mean ± SD (*n* = 4); * *p* < 0.01 and ** *p* < 0.001 vs. LPS (ANOVA with post hoc Dunnett tests).

**Figure 3 biomolecules-12-00091-f003:**
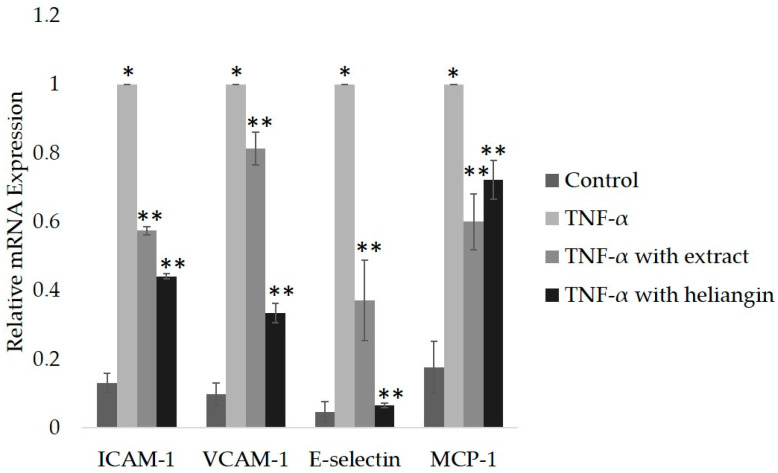
Effects of crude ethanol extract and heliangin on anti-inflammatory activity in VECs. Vascular endothelial cells were consecutively incubated with 200 µg/mL of crude ethanol extract (extract) or 15 µM heliangin for 24 h followed by incubation with 1 ng/mL TNF-α for 3 h. Levels of ICAM-1, VCAM-1, E-selectin, and MCP-1 mRNA expression were measured and are shown as the mean ± SD (*n* = 4); * *p* < 0.05 vs. control and ** *p* < 0.05 vs. TNF-α (ANOVA with Tukey’s tests).

**Figure 4 biomolecules-12-00091-f004:**
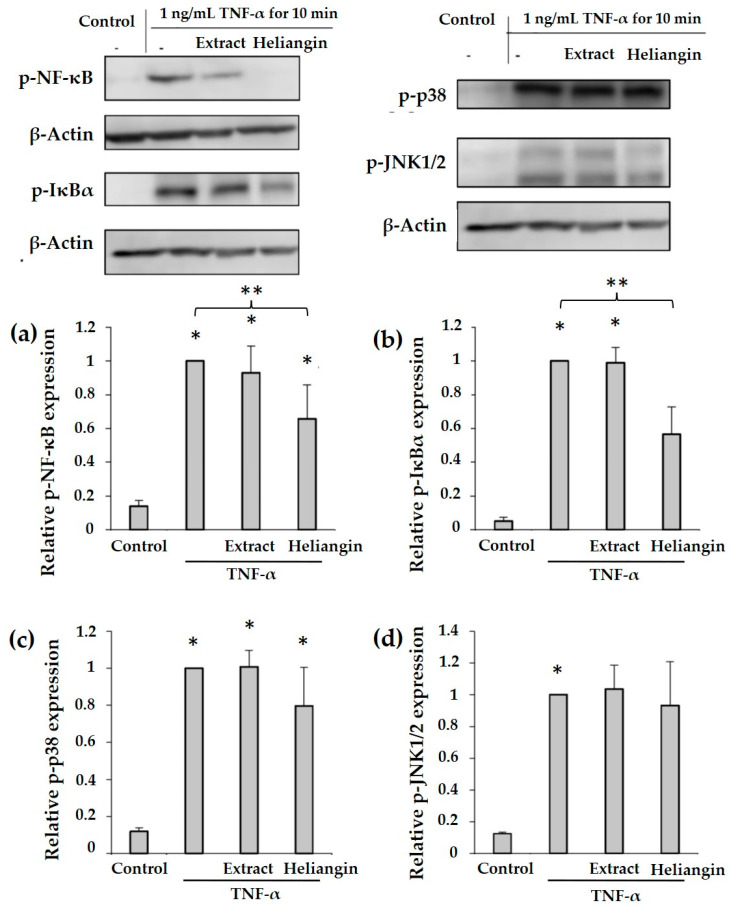
Ability of heliangin to inhibit NF-κB and MAPK pathway activation in VECs. Vascular endothelial cells were consecutively incubated with 200 µg/mL of crude ethanol extract (Extract) or 15 µM heliangin for 24 h and then 1 ng/mL TNF-α for 10 min. Relative protein expressions of (**a**) p-NF-κB, (**b**) p-IκBα, (**c**) p-p38 MAPK (p-p38), and (**d**) p-JNK1/2 were analyzed by Western blotting. Values are shown as the mean ± SD (*n* = 4); *p* < 0.05 vs. * control and ** TNF-α (ANOVA with Tukey’s tests).

**Table 1 biomolecules-12-00091-t001:** Primer sequences for real-time quantitative PCR.

Gene	Primer Sequences (5′–3′)
VCAM-1	F: CAAGGAAACGAAGAGTTTGGA
	R: TGTTGTCTTTACTGAGGGCTGAC
ICAM-1	F: TCAATGGAACCGAGAAGGAG
	R: GGAGGTGGGAAGCTGTAGAA
MCP-1	F: TCTCCAGTCACCTGCTFCTA
	R: TCCAGGTGGCTTATGGAGTC
E-selectin	F: AATGCCTTCAACCCAATGGA
	R: ACCGTCTTCAGGGTATCATGG
18S RNA	F: CGGCTACCACATCCAAGGAA
	R: CTCCAATGGATCCTCGTTAAAGG

**Table 2 biomolecules-12-00091-t002:** The ^1^H-NMR and ^13^C-NMR chemical shifts of fraction 6 (δ in ppm).

Carbon Number	Fraction 6	
	13C ^b^	1H ^b^
1	60.64 d	2.745 (dd, 10.3, 4.3, 1H)
2	32.59 t	1.675 (ddd, 10.3, 4.2, 2.1, 1H)
		2.40 (ddd, 14.9, 4.3, 4.3, 1H)
3	72.39 d	4.44 (S, 1H)
OH		2.25 (brs, 1H)
4	141.69 s	
5	126.57 d	5.26 (d, 10.9, 1H)
6	74.19 d	6.55(dd, 11, 1.5, 1H)
7	48.58 d	2.81 (S, 1H)
8	76.2 d	5.11 (S, 1H)
9	43.69 t	1.255 (dd, 14.9, 1.5, 1H)
		2.755 (dd, 14.8, 4.3, 1H)
10	58.63 S	
11	137.32 S	
12	169.58 S	
13	124.85 t	5.69 (d, 1.3, 1H)
		6.28 (d, 1.6, 1H)
14	19.75 q	1.4 (S, 3H)
15	23.02 q	1.75 (S, 3H)
1′	166.71 S	
2′	127.85 S	
3′	139.05 d	6.785 (dt, 6.8, 6.8, 1H)
4′	14.65 q	1.71 (d, 9.4, 3H)
5′	11.99 q	1.73 (S, 3H)

^b^ ppm.

## Data Availability

Not applicable.

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
