# Peer review of "Anti-Inflammatory Effects of Heliangin from Jerusalem Artichoke (Helianthus tuberosus) Leaves Might Prevent Atherosclerosis"

_biomolecules, 2022, doi:10.3390/biom12010091_

Round 1

Reviewer 1 Report

Dear authors.

Cardiovascular diseases are widespread all over the world. Despite many years of research, it is not possible to reduce mortality from these diseases. Any works that the results of which allow saving the patient's life is relevant. The topic touched upon in the article is relevant. The scientific content of the manuscript justifies its publication, but some additions and modifications will significantly improve the quality of the article.

Major comments:

1) The purpose of the work is not formulated. Abstract, Introduction and Conclusions forming different ideas about the purpose of the research. This needs to be fixed.

2) "LPS" must be defined at the first mention in the text.

3) Table 2, what do the letters next to the numeric values in this table?

4) Figure 1-5, the results of statistical analysis are not obvious. The authors should think about how to present these results (everything that is marked † is not unambiguous).

5) It is necessary to justify the use of different statistical tests for different indicators.

6) L.307-309, From the condition "... heliangin appeared to suppress genes signaling pathway"  does not follow "... that heliangin derived from Jerusalem artichoke leaves could help to prevent atherosclerosis". This can only become the hypothesis of the following studies

7) In the References, 26% of publications refer to 2017-2021 (the last 5 years); the remaining 74% of used sources are older than 5 years. It is recommended to increase the share of references to sources published over the last 5 years when analyzing the current state of research in the area under consideration, since this area of knowledge is rapidly developing.

Author Response

Dear Reviewer 1

Thank you very much for your comments. We revised and answered point to point as attached file. Please consider again

Best regards

Saiki Papawee (Correspondence)

Reviewer 2 Report

This manuscript describes the study about various biological activities of heliangin to elucidate the mode of action of its anti-inflammatory activity. Therefore, the difference between this study and the previous study should be clarified by mentioning the difference from ref20, the study on some biological activities of heliangin, not only in Introduction but also in Results and Discussion.

Please rewrite the English of the text to make it easier to understand. In particular, please make the connections between the individual sentences in the Introduction part a little clearer.

Because some previous works indicated heliangin as the main biologically active substance in the Jerusalem artichoke leaves, the part about identification of heliangin should be written simply. At least, it should not be necessary to show the data of Figures 1 and 2 in the main manuscript. It is better to show the structure of heliangin.

Please write the contents of Materials and Methods more carefully.

There are typos in Line 93, 216 and 217.

About Supplementary Information:

The NMR spectrum data is too small. HMQC and HMBC spectra data were not indicated even though it says that these spectra were measured in the Materials and Methods.

Author Response

Dear Reviewer 2

Thank you very much for your comments. We revised and answered point to point as attached file. Please consider again

Best regards,

Saiki Papawee (Correspondence)

Round 2

Reviewer 1 Report

Dear Authors

My comments are taken into account.

Author Response

Dear Reviewer1 

Thank you very much for your considering

Best regards

Saiki Papawee (Correspondence)

Reviewer 2 Report

Thank you for answering or correcting the parts which were pointed out in the last review.

About point 1:

The discussion of what was revealed in this study was well written in Results and Discussion as author mentioned. However, no comparison with the results of previous studies about heliangin was described in Results and Discussion. Therefore, it was difficult to understand how this study had advanced the heliangin science.

About point 3:

In previous studies that have identified heliangin as a major active compound in Jerusalem artichoke leaves, biological activity tests for screening have been performed as a matter of course during the isolation process. Therefore, in many cases, the biological test data of the separating process, which would have been carried out as a matter of course, is not included in the article about the study on the identification of the biologically active compound in Jerusalem artichoke leaves. Considering the importance of the data indicated in this manuscript, it would not be a good to show the data in Figures 1 and 2 in the main manuscript.

About point 4:

Although some parts in Materials and Methods have been improved, the data of the parameters on instrumental analysis or references of the biochemical experiments were not well described. Please write this section more carefully.

As some words and text used were not appropriate or difficult to understand, it would be better to ask to proofread.

About SI point 1:

Please check the line 213-215.

Author Response

Dear Reviewer 2

Thank you very much for your comments. We revised it as an attached file. The manuscript will show only 2nd revision as your advice. Please consider again

Best regard,

Saiki Papawee (Correspondence)
